# Mediverse: Multimodal Clinical Exploration and Search on a Single Graph

Andra Ionescu
KTH Royal Institute of Technology
Stockholm, Sweden
adio@kth.se

Paris Carbone
KTH Royal Institute of Technology
Stockholm, Sweden
parisc@kth.se

Sebastiaan Meijer
KTH Royal Institute of Technology
Stockholm, Sweden
smeijer@kth.se

Jayanth Raghothama
KTH Royal Institute of Technology
Stockholm, Sweden
jayanthr@kth.se

## ABSTRACT

Mediverse aims to bridge clinical and medical research and combine medical knowledge into a single graph that can be used to perform deep exploration and predictions across diverse modalities (e.g., genomic, clinical studies, medical, etc.). In particular, Mediverse intends to establish trust in medical data systems and exploit the benefits of graph representation learning methods, which are capable of addressing multiple key challenges that harm digitalisation in healthcare and medicine sectors today, namely: multi-modal data integration across clinics, complex search, and uncertainty-aware predictive computing.

**VLDB Workshop Reference Format:**
Andra Ionescu, Paris Carbone, Sebastiaan Meijer, and Jayanth Raghothama. Mediverse: Multimodal Clinical Exploration and Search on a Single Graph. VLDB 2026 Workshop: Biomedical Data Management Systems (BioDMS).

## 1 INTRODUCTION

Interoperability in healthcare has been developed from the bottom-up, layer by layer. It started with the development of exchange mechanisms (e.g., HL7 v2 [10]), which enabled hospitals to share structured clinical events. Next, terminology systems such as SNOMED-CT [7], LOINC [12], and much older ICD [14] were developed, providing standardised vocabularies to encode clinical concepts. However, even with shared transport and shared vocabularies, the structure of clinical information remained inconsistent. Information models, such as openEHR [19], responded to this gap, promising to solve interoperability by separating the technical layer (i.e., how data are stored) from the clinical layer (i.e., what the data tell us).

The model-first philosophy and the strict standards required significant upfront resources for modelling and governance, which only a few health systems could afford [1]. Because each layer was standardised independently, no unified information model was agreed upon. Each implementation imposed its own structure at the intersection of these standards, and we could see that hospitals

adopting the same standards still produced semantically incompatible datasets [9, 25]. This fragmentation emerged from the architecture of the bottom-up approach, a consequence that no single standard can undo.

Despite these flaws, the bottom-up approach contributed to solutions and advancements that solved technical and syntactic interoperability, which are two of the four levels of interoperability in healthcare: technical, syntactic, semantic, and organisational [17]. We are currently addressing the challenges of semantic interoperability, with the goal of enabling and solving organisational interoperability.

As such, we support a top-down approach, where rather than standardising exchange and hoping semantic agreement follows, the approach begins by defining a shared information model and derives storage, terminology, and exchange from it. Initiatives such as the OMOP Common Data Model (CDM) [24] represent partial implementations, providing a target for interoperability through data harmonisation.

Data harmonisation resolves heterogeneity along three dimensions [6]: syntactic, structural and semantic. With the syntactic dimension, different data formats are reconciled. The structural dimension addresses the different representations of the content. For example, information about a patient over three days is represented as a single row with `start_date` and `end_date` columns spanning three days, or as three rows with a `date` column indicating the date for each day. In the last dimension, data with the same meaning have to be harmonised. For example, data on young adults can be collected for different age groups (e.g., 18-25 years vs. 18-30 years) or it can have different meanings (e.g., young adults, teenagers, etc).

Currently, data harmonisation in healthcare is dominated by manual ETL pipelines that map the source data to a shared target schema and vocabulary, thus supporting the syntactic and structural dimensions. The persistent bottleneck is semantic alignment, the process of establishing correspondences between local terms and standardised vocabularies. This corresponds to the semantic dimension, which still requires human reviewers for validation.

In Mediverse, we embrace the semantic heterogeneity of healthcare, where clinical data vary between specialities, institutions, and regions. We address this diversity effectively using graph representation learning, leveraging medical ontologies and cross-ontology mappings to generate semantic links between disparate data fields, thus enabling a harmonised view over the fragmented datasets.

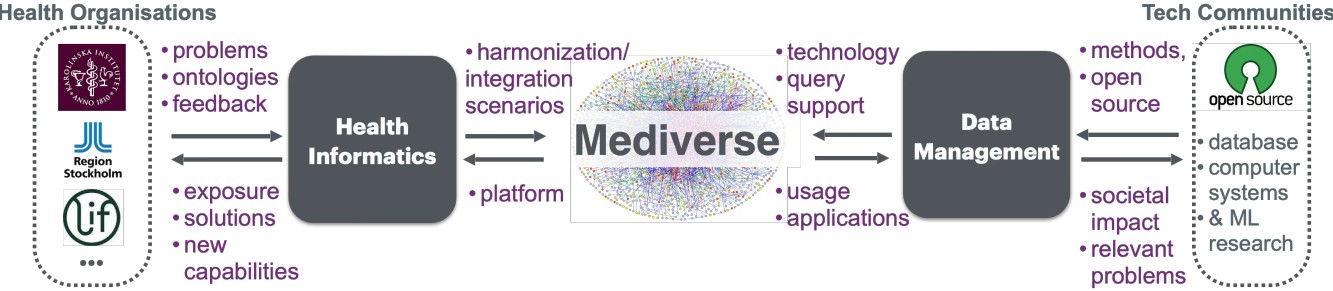

**Figure 1: Bridging communities: actors, tasks, benefits and interactions**

## 2 MEDIVERSE IN A NUTSHELL

Figure 1 encapsulates our vision of bridging two communities, health informatics and data management. Mediverse offers to health informatics a platform developed with data management support, exposure to solutions, technology, query systems, open-source methods, and databases. The data management community will be exposed to the problems and complexities of healthcare, get access to real datasets, and contribute to a direct societal impact by solving these problems. Moreover, clinicians contribute with domain knowledge and can offer the environment to deploy and test new solutions. As such, Mediverse aims to serve as a vehicle to bring the latest advancements in representation learning and knowledge graphs to the forefront of medical and clinical practice.

Our goal is to establish a reference platform for achieving data and metadata integration between healthcare providers and medical researchers. We leverage knowledge graphs for seamless exploration and interaction between semantically diverse elements, such as genomic sequences, clinical and medical imaging records. Graph neural networks (GNNs) extend this by capturing multi-hop reasoning paths that reflect causal and taxonomic relationships in clinical ontologies [11, 20]. With GNNs, we showcase the capabilities of predictive exploration in healthcare, revealing insights that would otherwise remain hidden. GNN-powered inference will equip clinicians with the ability to anticipate patient outcomes, outline potential treatment options, and accelerate medical discovery. Mediverse will further incorporate black-box uncertainty estimation methods in graph representation learning tasks, to address accuracy, which remains a crucial barrier to applying inference and neural networks in healthcare. This will ensure that clinicians and researchers can rank predicted outputs using statistically accurate uncertainty bounds.

Furthermore, by democratising the use of responsible AI in healthcare, we aim to attract the attention of clinicians and biomedical researchers to emerging technologies, eliminating data integration concerns that typically take years to solve, while enhancing digitalisation in healthcare, productivity, and trust by shifting focus to the healthcare problems that matter.

## 3 MULTI-MODAL INTEGRATION

There are many standards that facilitate both the organisation of healthcare data (openEHR, SNOMED-CT, LOINC), as well as their exchange (i2b2, HL7 FHIR) [9, 29]. However, the heterogeneity of

these standards and their partial coverage across sites and institutions [9] is one of the core challenges for data harmonisation. Figure 2 illustrates how common concepts across different domains, or across different ontologies of the same domain, can be linked with each other.

With this project, we aim to get a step closer to solving semantic interoperability, which is defined as "the ability of two or more systems or components to exchange information and to use the information that has been exchanged" [9]. As such, we work on the core component of semantic interoperability, semantic harmonisation, defined as "the process of combining multiple sources and representations of data into a form where items of data share meaning" [8]. Unlike conventional approaches, we investigate semantic harmonisation *without* transforming the data into a common format, since doing so is prone to information loss and requires a predefined target schema. Instead, to enable meaningful integration across diverse knowledge domains, representation formats, and data models, Mediverse leverages knowledge graphs to create flexible, standard-agnostic interoperability approaches.

We achieve this by developing a single graph representation of (medical) ontologies. These ontologies are open-source, many are available as graphs, and nearly all of them have bindings (i.e., links between concepts across ontologies) with each other. These representations and bindings are imported into a graph database to provide a single knowledge base of medical/clinical concepts. As illustrated in Figure 2, we want to query data using a single query language (e.g., Cypher) and the expressiveness of graph databases.

Our first use case connects Electronic Health Records (EHR) data with ontologies. We propose a semi-automatic approach for concept normalisation, a sub-problem of semantic alignment, which links individual data values from EHR to concepts in a standardised vocabulary. EHR data comes in multiple formats, the most common being free text (i.e., medical notes) and tabular values (i.e., tables with patient records, admissions, prescriptions, etc). Semantic alignment in EHR data requires integrating contextual, structural, and domain-specific signals to ensure correct interpretation across heterogeneous data sources. Thus, the type of EHR data determines the approaches we need for integration. Free text carries richer context, thus we apply biomedical entity linking solutions. Tabular values carry column/row context, thus we apply standard schema matching techniques [21] alongside solutions from entity reconciliation. Once the entity boundaries are known, we perform concept normalisation.

## Ontological Data Harmonisation

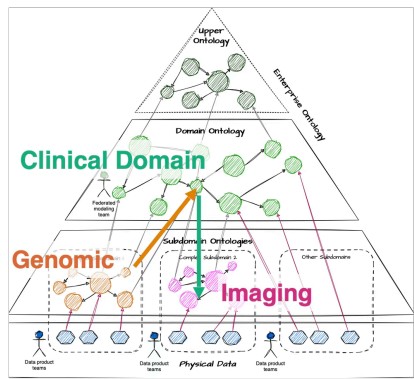

```
MATCH (patient:Patient)-[:HAS_GENE]
    ->(gene:GenomicFeature { ID:'MGMT'})
MATCH (gene)-[:ASSOCIATED_WITH]->(effect:DrugResistance)
MATCH (patient)-[:RECEIVED_TREATMENT]
    ->(therapy:Treatment { name: 'Targeted Therapy'})
MATCH (patient)-[:UNDERWENT_IMAGING]->(mri:MRI_Scan)
WHERE mri.result = 'Tumor Shrinkage'
```

**Figure 2: Harmonisation using a single graph of multi-modal data and a sample Cypher query**

## Multi-modal Graph Query Search

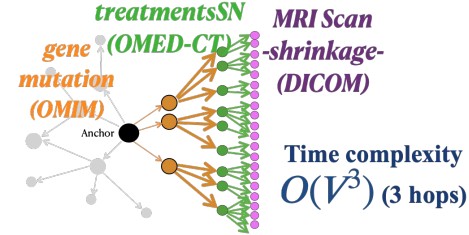

**Figure 3: Graph query search and predictive execution**

## Predictive Search via Representation-Learning (Query2Box)

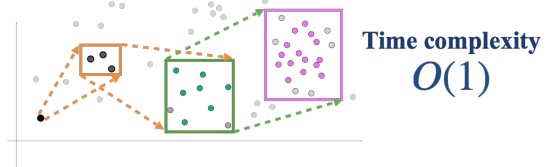

## 4 UNCERTAINTY AWARE SEARCH AND EXPLORATION

Graphs are highly generalised models for data and their relationships. In the medical sector, a graph can represent drug and patient vertices connected with respective edges representing interactions (drug-drug) or reactions (patient-drug). Other graphs can connect diseases, genetic mutations, or patients with clinical trials and outcomes. When graphs grow, they tend to follow certain trends such as becoming sparse (i.e., with missing connections) and following a typical power law distribution of connections. In graph theory, these are also known as natural graphs.

Traversing natural graphs to detect certain patterns is problematic due to the disproportional amount of computing power necessary to reach abundant information, a corresponding search space also known to follow "non-Euclidean" geometry. To address these issues, the field of graph representation learning studies techniques that can efficiently compress graph information into fixed, lower-dimensional Euclidean spaces such as embeddings [13]. More particularly, GNNs have already showcased breakthroughs in fields such as protein interaction [18] and physics simulations [28]. Embeddings are also shown to be capable of learning computational relations within graphs.

We reframe semantic harmonisation as a graph alignment problem rather than a schema transformation one. We will experiment with both supervised and unsupervised methods, such as EMBERT (Entity-rich Medical BERT) [3], FINAL (Fast Attributed Network Alignment) [30], SEU (Simple but Effective Unsupervised entity alignment) [23], CONE-Align [5], investigate and showcase the domain-efficacy of these models with a focus on multimodal medical/clinical data.

Figure 2 shows a 3-hop Cypher query that searches through genomic, pharmacological, and clinical ontologies. On a medical dataset, such as AIMedGraph [26] which has millions of connections, this computation could take up to a minute, since the algorithmic graph search has $O(V \times h)$ complexity (where $V$ is the number of vertices, and $h$ represents the steps/hops). In contrast, using Query2Box [27] on fixed dimensions reduces this to $O(1)$, drastically speeding up the process to second, or sub-second latency, while also detecting hidden links in the data. Figure 3 illustrates the search complexity using algorithmic graph traversals compared to representation learning.

Predictability in query processing was explored in the context of Approximate Query Processing (AQP) [4], an early attempt based on data sampling to speed up query responses in SQL databases at the expense of inaccurate output. BlinkDB [2] is a known system that featured error bounds but could not achieve widespread adoption due to its inability to generate new missing connections in the data and diminishing returns in speed over high accuracy penalties. More recent research in data management explores the use of ML [22], yet, their focus is on optimisation purposes, such as learning indices, and not to enrich data exploration.

Within an interoperability pipeline, conformalised link prediction can offer a concrete mechanism to quantify and user-bound uncertainties. Systems such as OrbDB [15] and confidence calibrated vector search engines (e.g. Conann [16]) can use conformal scores to recalibrate similarity results and predictions so that the reported confidence matches observed correctness. Thus, we will explore the use of black-box uncertainty estimators (e.g., conformal and Venn predictors) to estimate uncertainties with statistical bounds on graph representation learning models. The system will estimate and display the uncertainty of the computed outputs, along with the average uncertainty of the entire query.

# 5 CONCLUSION

Mediverse addresses a long-standing barrier in semantic interoperability by reframing harmonisation as a graph alignment problem rather than a schema transformation one. Through a unified graph representation of medical ontologies and open clinical datasets, combined with uncertainty-aware graph learning, the project lays the groundwork for interoperability approaches that are flexible, standard-agnostic, and robust under ambiguity. Ultimately, Mediverse aspires to make heterogeneous clinical data meaningfully usable across institutions, unlocking new opportunities for both primary care and secondary research.

# 6 AUTHORS

**Andra Ionescu** is a Postdoctoral Researcher at KTH Royal Institute of Technology, in the Department of Biomedical Engineering and Health Systems. Her current work focuses on graph-based clinical data harmonisation and semantic interoperability challenges.

**Paris Carbone** is Associate Professor at KTH Royal Institute of Technology and director of the Data Systems Lab. He is working on scalable data processing systems, generative computing, data streaming, and graph databases. He is a co-recipient of the ACM SIGMOD Systems Award 2023 (Apache Flink).

**Sebastiaan Meijer** is Professor of Health Care Logistics at KTH Royal Institute of Technology. He is working on healthcare, health prevention and promotion systems, and is the coordinator of the European Digital Innovation Hub Health Data Sweden.

**Jayanth Raghothama** is Associate Professor at KTH Royal Institute of Technology, in the Department of Biomedical Engineering and Health Systems. His research interests include complex systems, health informatics, policy design, and applied ML.

# ACKNOWLEDGMENTS

This work was partially funded by Digital Futures.

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
