# OpenReview forum: "Mediverse: Multimodal Clinical Exploration and Search on a Single Graph"
_VLDB.org/2026/Workshop/BioDMS — BioDMS 2026 LightningTalk_

### Official Review · Reviewer_Mbpq · 2026-06-01

**Summary:**

The authors propose a project called Mediverse, a single graph to perform deep exploration and predictions across diverse modalities, based on graph representation learning.

**Confidence Of Review:**

3

**Detailed Feedback Points:**

S1. The integration and harmonisation of medical data of multiple modalities is an important research direction with huge potential practical impact.

S2. The project is very ambitious.

W1. The paper is hard to follow, since it presents a very broad and high-level proposal, and introduces a lot of terminology and claims, which are hard to verify, e.g., that "In practice, technical and syntactic interoperability have been solved". I am missing a concrete technical example that outlines what problems are not solved by current methods and how Mediverse would solve them.

W2. Several technical details are unclear, the paper states that the authors investigate "semantic harmonisation without transforming the data into a common format", but then postulates that this can be achieved by "developing a single graph representation of medical ontologies", isn't that a common format? The choice of graph representation learning and particular search strategies (like Query2Box) also seem rather arbitray, it would be good to show some mini-experiment to outline their benefits.

W3. The paper proposes a data management heavy work plan, what is the role of medical researchers here?

**Relevance For Biodms:**

2

---

### Official Review · Reviewer_Pc7f · 2026-06-10

**Summary:**

Mediverse suggest a graph database representation of data instead of harmonizing. They suggest mapping each data entity to a node of a knowledge graph in that nodes represent ontology items; they developed this graph mapping ontologies. Further, they suggest using graph embedding techniques to make queries faster compared to algorithmic graph traversal; to mitigate the subsequent imprecision, they apply uncertainty estimation techniques to ensure the user does get back results with at most the uncertainty they want.

**Confidence Of Review:**

1

**Detailed Feedback Points:**

Pros
- the idea of saving data for common access and analysis in a joint graph database is a promising approach
- combining this with graph embeddings and particularly an uncertainty assessment to make retrieval faster is well needed in this case

Cons
- data items are mapped applying some conversion tool; this means loss of semantic meaning will still occur via mapping
- chapter 2 seems very only like selling, without clear, provable claims; especially how data is going to be available to researchers is not clearly addressed (considering data protection standards etc).
- whether the entire platform will be open source is rather unsure (just open-source methods are mentioned); for the community and supporting research, this is an issue

**Relevance For Biodms:**

2

---

### Official Review · Reviewer_vDZ6 · 2026-06-16

**Summary:**

The submission presents Mediverse, a project aiming to advance data integration/semantic interoperability by building a graph integrating various medical ontologies, based on a knowledge graph. Additionally, it envisages GNN-based semantic search capabilities with uncertainty-awareness/conformal prediction.

**Confidence Of Review:**

2

**Detailed Feedback Points:**

Strong:
- the push towards better data integration in medical/health informatics is definitely needed, and new approaches are necessary
- conformal predictions are a really nice idea, especially in this context

Opportunities for improvement:
- presentation would need to be a bit clearer. It's not fully clear whether Mediverse is a system, a graph, a database, an application, a set of search methods... Is the goal more on the method developoment side or building a data integration tool, or a user-facing system. Will Mediverse be a learned ML model, or an explicitly coded software system? What is the current status of the work? In the current state of the write-up, it is challenging to judge the relative contribution of the project.
- focus seems to be mode medical informatics than strictly biomedical research, but it most likely not a problem, and similar methods could be useful for data integration across ontologies in molecular/computational biology. Maybe this could also be an interesting direction?
- the authors write Mediverse reframes harmonisation as a graph alignment problem, but it is not fully clear how exactly is this a graph alignment problem, rather than merging all ontologies into one big graph with subgraphs corresponding to individual ontologies. Could the authors clarify which graphs are being aligned, with which graph alignment algorithm?
- It does not sound feasible that Mediverse would be able to combine "*all* medical knowledge into a single graph", as written by the authors. Even if all the ontologies across the world were to be integrated, along all the digitally available medical literature, this would still not constitute all medical knowledge. Mediverse can still be useful even if it doesn't capture all of medical knowledge, but extreme claims should be avoided unless there's a strong evidence.
- there's potential for an interesting paper, although probably not at a biomedical journal, but rather an interdisciplinary one catering to the medical/health informatics community

**Relevance For Biodms:**

2